# Chronic Inflammatory Enteropathy and Low-Grade Intestinal T-Cell Lymphoma Are Associated with Altered Microbial Tryptophan Catabolism in Cats

**DOI:** 10.3390/ani14010067

**Published:** 2023-12-23

**Authors:** Patrick C. Barko, David A. Williams, Yu-An Wu, Joerg M. Steiner, Jan S. Suchodolski, Arnon Gal, Sina Marsilio

**Affiliations:** 1Departments of Veterinary Clinical Medicine and Pathobiology, University of Illinois at Urbana-Champaign, Urbana, IL 61802, USA; 2Gastrointestinal Laboratory, School of Veterinary Medicine and Biomedical Sciences, Texas A&M University, College Station, TX 77843, USA; 3Department of Veterinary Clinical Medicine, University of Illinois at Urbana-Champaign, Urbana, IL 61802, USA; 4Department of Veterinary Medicine and Epidemiology, UC Davis School of Veterinary Medicine, Davis, CA 95616, USA

**Keywords:** microbial indole catabolites of tryptophan, indole, chronic enteropathy, inflammatory bowel disease, alimentary small cell lymphoma

## Abstract

**Simple Summary:**

Chronic inflammatory enteropathy (CIE) and low-grade intestinal T-cell lymphoma (LGITL) are common chronic intestinal disorders in cats. Gut bacteria are implicated in the initiation and progression of chronic intestinal disorders, and changes in the composition of gut bacteria have been associated with CIE and LGITL in cats. Microbial indole catabolites of tryptophan (MICT) are chemicals produced by gut bacteria that support intestinal health. We hypothesized that, compared with healthy cats, blood concentrations of MICTs would be decreased in cats with CIE and LGITL. Using archived blood samples, we measured tryptophan and eleven of its derivatives, including eight MICTs, and compared them among cats with CIE, LGITL, and healthy controls. Consistent with our hypothesis, the concentrations of tryptophan and five different MICTs (indolepropionate, indoleacrylate, indolealdehyde, indolepyruvate, indolelactate) were decreased in cats with CIE and LGITL compared with those in healthy controls. Our findings are likely explained by changes in tryptophan metabolism related to disturbances in gut bacterial communities and intestinal inflammation and in cats with CIE and LGITL. These findings suggest that MICTs are promising biomarkers that can be used to understand how intestinal bacteria contribute to chronic intestinal disorders such as CIE and LGITL in cats.

**Abstract:**

Chronic inflammatory enteropathy (CIE) and low-grade intestinal T-cell lymphoma (LGITL) are common chronic enteropathies (CE) in cats. Enteric microbiota dysbiosis is implicated in the pathogenesis of CE; however, the mechanisms of host–microbiome interactions are poorly understood in cats. Microbial indole catabolites of tryptophan (MICT) are gut bacterial catabolites of tryptophan that are hypothesized to regulate intestinal inflammation and mucosal barrier function. MICTs are decreased in the sera of humans with inflammatory bowel disease and previous studies identified altered tryptophan metabolism in cats with CE. We sought to determine whether MICTs were decreased in cats with CE using archived serum samples from cats with CIE (*n* = 44) or LGITL (*n* = 31) and healthy controls (*n* = 26). Quantitative LC-MS/MS was used to measure serum concentrations of tryptophan, its endogenous catabolites (kynurenine, kynurenate, serotonin) and MICTs (indolepyruvate, indolealdehyde, indoleacrylate, indoleacetamide, indoleacetate, indolelactate, indolepropionate, tryptamine). Serum concentrations of tryptophan, indolepropionate, indoleacrylate, indolealdehyde, indolepyruvate, indolelactate were significantly decreased in the CIE and LGITL groups compared to those in healthy controls. Indolelactate concentrations were significantly lower in cats with LGITL compared to CIE (*p* = 0.006). Significant correlations were detected among serum MICTs and cobalamin, folate, fPLI, and fTLI. Our findings suggest that MICTs are promising biomarkers to investigate the role of gut bacteria in the pathobiology of chronic enteropathies in cats.

## 1. Introduction

Chronic inflammatory enteropathy (CIE), also called inflammatory bowel disease (IBD), and low-grade intestinal T-cell lymphoma (LGITL) are two common forms of chronic enteropathy (CE) in cats [1,2,3]. The coexistence of inflammatory and neoplastic lesions in cats with CE and the progression of CIE to LGITL in some affected cats suggest that these two conditions share a common etiopathogenesis [1,2]. Complex and reciprocal interactions among commensal microbes and host tissues are critical for regulating inflammatory responses and cell proliferation in the enteric mucosa, and these interactions are largely mediated by bacterial metabolites, including short chain fatty acids, secondary bile acids, and tryptophan derivatives [3,4]. Conversely, enteric microbiota dysbiosis (EMD), defined by an imbalance between beneficial commensal microbes and opportunistic pathobionts, contributes to the pathogeneses of inflammatory and neoplastic disorders of the intestinal mucosa [5]. EMD has been identified in cats with CIE and LGITL, but there is a paucity of data regarding the mechanisms through which enteric microbiota influence mucosal health and contribute to the pathogenesis of CE in cats [6,7].

Tryptophan is an essential amino acid and emerging evidence from research in humans and rodents suggests that catabolic derivatives of tryptophan participate in regulating intestinal mucosal health and homeostasis. Tryptophan is catabolized through endogenous and exogenous (microbial) pathways (Figure 1A) [8,9,10,11,12]. Approximately 90–95% of tryptophan is catabolized through the endogenous kynurenine pathway (KP), which is controlled by the rate-limiting enzymes tryptophan 2,3-dioxygenase (TDO) in the liver and indoleamine 2,3-dioxygenase-1 (IDO1) in extrahepatic tissues, including leukocytes and intestinal epithelial cells [8]. The TDO-dependent hepatic KP is primarily regulated by endocrine factors (e.g., glucocorticoids, glucagon), whereas the IDO-dependent extrahepatic KP is stimulated by inflammatory cytokines [13]. Approximately 5% of ingested tryptophan escapes absorption in the small intestine and becomes a substrate for metabolism by gut microbiota, resulting in the generation of microbial indole catabolites of tryptophan (MICT), which are readily absorbed by the jejunal, ileal, and colonic mucosa [12,14,15]. MICTs, including indolepropionate, indolealdehyde, indolelactate, and indoleacrylate, are exclusive products of gut microbial metabolism [15,16,17]. Smaller fractions of ingested tryptophan (1–2%) are used for the synthesis of serotonin by enterochromaffin cells, and tryptamine, which can be produced by both bacterial and mammalian cells [18,19,20]. Emerging evidence suggests that disruptions in tryptophan catabolism are associated with inflammatory bowel diseases and that MICTs play a role in regulating gastrointestinal immunity and the mucosal barrier function [12,15,21].

Recent studies have revealed altered tryptophan catabolism in humans with IBD, characterized by decreased serum concentrations of tryptophan and indolepropionate, decreased fecal concentrations of indoleacetate, and increased serum concentrations of kynurenine pathway catabolites [22,23,24,25,26]. Serum tryptophan concentrations were inversely correlated with disease activity and serum C-reactive protein, whereas increasing ratios of kynurenine–tryptophan were associated with increased odds of observing endoscopic inflammatory lesions and worse clinical outcomes in humans with IBD [23,24]. Mechanistic investigations in rodents and in vitro with human intestinal epithelial cell cultures implicate MICTs in regulating intestinal mucosal inflammation and mucosal barrier function (Figure 1B). MICTs induce aryl hydrocarbon (AhR) and pregnane-X (PXR) receptor-mediated expression of tight junction proteins (TJP), mucin, anti-microbial peptides (AMP), and anti-inflammatory cytokines (e.g., IL-10) while decreasing the expression of pro-inflammatory cytokines (e.g., TNFα) in the enteric mucosa. [12,15,20,27,28,29,30,31]. A variety of bacteria are known to catabolize tryptophan into MICTs, including various species of *Lactobacillus*, *Bacteroides*, *Bifidobacterium*, *Peptostreprococcus*, and *Clostridium*, which are often decreased in EMD associated with chronic enteropathies in humans and other animal species, including cats [15,30]. Not only are MICTs potential biomarkers for disruptions in gut microbial metabolism, but they are also promising therapeutic targets. For example, the administration of probiotics containing MICT-generating species and/or dietary MICT supplementation ameliorates mucosal inflammation and EMD in animal models of IBD [30,32,33].

A previous untargeted metabolomics study of feces from cats with CE identified evidence of altered tryptophan metabolism in cats with CE, and a pilot study revealed decreased concentrations of tryptophan and indole-3-lactic acid in the sera of cats with CIE and LGITL compared with healthy controls [34,35]. We hypothesized that CIE and LGITL would be associated with altered tryptophan catabolism in cats, characterized by decreased serum concentrations of tryptophan and MICTs, and increased serum concentrations of kynurenine (Figure 1C). Here, we report the results of a comprehensive survey of tryptophan catabolites in the sera of cats with CE and healthy controls using quantitative LC-MS/MS. By investigating associations between microbial tryptophan catabolites and CIE and LGITL, we aim to generate novel insights that can be used to understand the role of gut microbiota in the pathogenesis and pathophysiology of CE in cats.

## 2. Materials and Methods

### 2.1. Patient Population and Sample Acquisition

This was a case–control investigation using archived serum samples from client-owned cats with spontaneously occurring CIE and LGITL. Samples from cats with CE were collected between 2015 and 2019 at three different institutions for either routine diagnostics or from participants in previous investigations unrelated to this study. These cats had undergone comprehensive diagnostic evaluation by their primary veterinarians and/or the clinical investigators. The medical records and experimental metadata for these cats were reviewed by the authors to extract data related to clinical signs, physical examination findings, clinicopathologic results (including serum concentrations of cobalamin, folate, fPLI, and fTLI), results of histopathology, medication history, and other concurrent diagnoses. In cases with missing values for serum concentrations of cobalamin, folate, fPLI, and fTLI, these assays were performed on residual serum, when available. Though diagnostic evaluations varied among institutions, we applied uniform inclusion and exclusion criteria. To be eligible for inclusion, cats had to have documentation of active clinical signs of gastrointestinal dysfunction, including vomiting, diarrhea, weight loss, anorexia, or some combination thereof, that had persisted for more than three weeks. Additionally, all cats with clinical signs consistent with CE had to have histopathologic findings consistent with CIE or LGITL based on examination of endoscopic or full-thickness, surgical biopsies by a board-certified veterinary pathologist. For cases in which the pathologist could not differentiate between CIE and LGITL based on routine histopathology, ancillary testing with immunohistochemistry and PCR for antigen receptor rearrangement (PARR) was performed for diagnostic confirmation. A final diagnosis of CIE or LGITL was reached upon integration of results from histopathology with immunohistochemistry and PARR when available, consistent with previous studies in cats and current EuroClonality/BIOMED-2 guidelines and a recent consensus statement from the American College of Veterinary Internal Medicine (ACVIM) [36,37,38,39,40,41,42]. Cats were excluded if they were documented to have received antibiotics or immunomodulatory drugs within 4 weeks of sample collection. Cats with documented hyperthyroidism were excluded unless clinical signs of gastrointestinal dysfunction persisted following the treatment of hyperthyroidism. Cats with documented diagnoses of other neoplasms (aside from LGITL) or exocrine pancreatic insufficiency (serum fTLI concentrations ≤ 12.0 µg/L) were excluded.

Archived serum samples from healthy cats collected for previous, unrelated investigations between 2015 and 2021 were used as healthy controls. The health status of cats in this group was verified by medical histories covering the following areas: attitude/activity, appetite, drinking, urination, chronic illnesses, weight loss, vomiting, diarrhea, and treatment with antibiotics or immunomodulatory drugs. Physical examinations were performed by board-certified small animal internal medicine specialists. The body condition scores were assessed using a previously established nine-point scoring system [43]. Blood was collected from a peripheral vein or the jugular vein, and the following tests were performed: complete blood count, serum chemistry profile, and serum concentrations of total T4, cobalamin, folate, feline pancreatic lipase immunoreactivity (Idexx Spec fPL), and feline trypsin-like immunoreactivity (fTLI; TAMU). Cats with gastrointestinal signs (weight loss, hyporexia, vomiting > 2x/month, diarrhea) within 6 months prior to enrollment were excluded. Cats with systemic diseases, chronic illnesses, or clinically significant laboratory abnormalities were also excluded from the study. Cats with serum concentrations of cobalamin < 290 ng/L, folate < 9.7 µg/L or > 21.6 µg/L, fPLI > 3.5 µg/L, or fTLI ≤ 12.0 µg/L were excluded. Finally, cats that had received any antibiotics or immunomodulatory drugs within 6 months prior to sample collection were excluded.

Archived sera from the cats with CE and healthy controls had been stored below −70 °C at participating institutions and mailed overnight on dry ice to the investigators for use in the present study. Following receipt, they were stored at −80 °C and transported on dry ice for measurement of serum tryptophan catabolites.

### 2.2. Quantification of Tryptophan Catabolites

Quantitative (targeted) liquid chromatography–mass spectrometry (LC-MS) was used to measure serum concentrations of tryptophan (Sigma-Aldrich Cat. PHR1176), seven different MICTs, and other tryptophan catabolites, including kynurenine, kynurenic acid, tryptamine, and serotonin against known dilutions of analytic standards. All analytic standards were sourced from the same chemical supplier (Sigma-Aldrich, St. Louis, MO, USA) at the highest purity available: tryptophan (Cat. PHR1176), indolepyruvate acid (Cat. I7017), indolealdehyde (Cat. 129445), indoleacrylate (Cat. I2273), indoleacetamide (Cat. 286281), indoleacetate (Cat. I3750), indolelactate (Cat. I5508), indolepropionate (Cat. 220027), kynurenine (Cat. 67653), kynurenic acid (Cat. 67667), tryptamine (Cat. 76706), and serotonin (Cat. 14927). The analytic standards were diluted in 70% methanol and the dilution range for the calibration curve was 0.25–5000 ng/mL. LC-MS/MS was performed at the Carver Metabolomics Core in the Roy J. Carver Biotechnology Center (University of Illinois, Urbana, IL, USA). Serum samples (30 µL) were spiked with 10 µL of internal standards (1 µg/mL), deproteinized with methanol (70 µL), centrifuged, and 2 μL of the supernatant was injected into the LC-MS. Chromatography was performed using the Vanquish system (Thermo Scientific, Waltham, MA, USA), with a Waters Acquity UPLC BEH C18, (2.1 × 150 mm; 1.7 μm) column with a flow rate of 300 μL/min and two mobile phases (0.1% formic acid in water; 0.1% formic acid in acetonitrile) with a column chamber temperature of 40 °C. Mass spectrometry utilized a TSQ Altis LC-MS/MS system (Thermo Scientific). Data were acquired in both positive and negative SRM modes at 3500 V and 5000 V, respectively. Peak integration and quantitation were performed with Thermo TraceFinder software (version 4.1).

### 2.3. Statistical Analysis

Statistical analyses were performed using the R language for statistical computing (v. 4.2.1) [44]. Reproductive status was compared among the groups using the Chi-square (χ^2^) test. The distribution of continuous numerical variables was assessed by examining histograms and Shapiro–Wilk tests. Owing to violations of normality for nearly all analytes, non-parametric methods were utilized for inferential statistics and the results are described using the median and interquartile range (IQR). Kruskal–Wallis tests were used to compare numerical variables among the groups. Variables that varied significantly in the overall test were compared group-wise using post hoc Dunn’s tests. To detect correlations among gastrointestinal biomarkers and serum concentrations of tryptophan catabolites, Spearman’s rank correlation coefficients and corresponding *p*-values were calculated for all pairwise combination of variables. As the Kruskal–Wallis and Spearman rank correlation tests utilized multiple comparisons, *p*-values were adjusted (*p*_adj_) using the Benjamini–Hochberg method to control for false discovery [45]. Features were considered significantly different using a two-tailed significance threshold of α < 0.05. 

## 3. Results

### 3.1. Patient Population

Archived sera from 101 cats were obtained, including 44 cats with CIE, 31 cats with LGITL, and 26 healthy controls. Demographic and clinical characteristics of this patient cohort are summarized in Table 1. There were significant differences in age (*p* = 0.01) and body condition score (*p* < 0.001) among the groups. Cats in the LGITL group (median 12 years, IQR = 8.0–13.0; *p* < 0.001) were significantly older than healthy controls (median 10 years, IQR = 8–11), but there were no other statistically significant differences in age among the groups. Compared with healthy controls (median 6, IQR = 5–8), cats in the CIE (median 5, IQR = 4–5; *p* < 0.001) and LGITL groups (median 4, IQR = 4–5; *p* < 0.001) had significantly lower body condition scores (1–9 scale), but there were no significant differences between the CIE and LGITL groups (*p* = 0.39). All cats were either spayed females or neutered males and there were no statistically significant differences in the proportions of spayed females or neutered male cats among the groups (*p* = 0.853). Serum concentrations of cobalamin (*p* < 0.001) and fTLI (*p* < 0.001) were significantly different among the groups and were significantly lower in cats with CIE (median 743.0 ng/L, IQR = 378.8–891.3; *p* < 0.001) and LGITL (median 615 ng/L, IQR = 209.0–906.5; *p* < 0.001) compared with healthy controls (median 1000 ng/L, IQR = 826.3–1000.0). There was no significant difference in serum cobalamin concentrations between the CIE and LGITL groups (*p* = 0.50). Serum fTLI concentrations were significantly higher in cats with LGITL (median 56.8 µg/L, IQR = 38.5–84.4) than the CIE (median 32.2 µg/L, IQR = 27.8–47.5; *p* = 0.003) and healthy control groups (median 32.6 µg/L, IQR = 23.1–39.0; *p* < 0.001), but there were no other significant differences between the CIE and LGITL groups. Serum fPLI concentrations were significantly different among the groups (*p* = 0.045) in the Kruskal–Wallis test and were higher in the LGITL group compared with the CIE and healthy control groups, but the differences were not statistically significant in the pairwise Dunn’s tests. There were no significant differences in serum folate concentrations among the groups (*p* = 0.66). Extra-intestinal comorbidities in the CE groups included chronic renal disease (*n* = 17) and hyperthyroidism (*n* = 6), with two of these cats having both chronic renal disease and hyperthyroidism. Histopathologic findings from the livers and pancreata of 57 cats with CE were available for review in the medical records. Inflammatory infiltrates were identified in the livers of 11 cats and in the pancreata of 3 cats, and all 3 cats with pancreatitis also had inflammatory infiltrates present in the liver.

### 3.2. Serum Concentrations of Tryptophan Catabolites

Serum concentrations of tryptophan (*p* < 0.001) varied significantly among the groups (Table 2; Figure 2) and were lower in the CIE (median 10.8 µg/mL, IQR = 9.0–13.3; µg/mL; *p* < 0.001) and LGITL (median 9.3 µg/mL, IQR = 8.4–11.7; µg/mL; *p* < 0.001) groups compared with those in the healthy controls (median 16.9 µg/mL, IQR = 15.5–18.7). There were no significant differences in serum tryptophan concentrations between the CIE and LGITL groups (*p* = 0.17). Serum concentrations of kynurenine (*p* = 0.23) and kynurenate (*p* = 0.63) did not vary significantly among the groups, despite being higher in the CIE and LGITL groups compared with those in the healthy controls. There were no significant differences in serum serotonin concentrations (*p* = 0.15) among the groups.

Concentrations of several serum MICTs varied significantly among the groups (Table 2; Figure 2): indolepropionate (*p* < 0.001), indoleacrylate (*p* < 0.001), indolelactate (*p* < 0.001), indolepyruvate (*p* = 0.001), and indolealdehyde (*p* < 0.001). Indolacetamide concentrations were found to be significantly different (*p* = 0.034) among the groups in the overall test; however, the medians and IQRs did not differ among the groups and there were several extreme outliers. Thus, indolacetamide was not considered significantly different and was not subjected to post hoc testing. Post hoc pairwise comparisons between groups revealed that serum concentrations of indolepropionate, indoleacrylate, indolepyruvate, and indolealdehyde were significantly decreased in the CIE and LGITL groups compared with healthy controls (Table 3). Serum indolelactate concentrations were significantly lower (*p*_adj_ = 0.008) in cats with LGITL (median 105.0 ng/mL, IQR = 79.0–145.2) compared with IBD (median 165.5 ng/mL, IQR = 102.9–271.8), but there were no significant differences in the serum concentrations of other MICTs between the CIE and LGITL groups. Serum concentrations of tryptamine were not significantly different among the groups (*p* = 0.16). Graphical outputs (boxplots) for all analytes are shown in the Appendix A.

### 3.3. Correlation Analysis

Several serum tryptophan catabolite concentrations were significantly correlated with serum cobalamin, folate, fTLI and fPLI concentrations (Table 4). Serum cobalamin concentrations were positively correlated with indolealdehyde (r = 0.39; *p*_adj_ < 0.001), tryptophan (r = 0.39; *p*_adj_ = 0.001), indoleacrylate (r = 0.37; *p*_adj_ = 0.001), indolepyruvate (r = 0.31; *p*_adj_ = 0.007), and indolelactate (r = 0.27; *p*_adj_ = 0.02); and negatively correlated with kynurenine (r = −0.25; *p*_adj_ = 0.03). Serum fPLI concentrations were correlated positively with kynurenine (r = 0.28; *p*_adj_ = 0.016) and negatively with tryptophan (r = −0.28; *p*_adj_ = 0.016), indoleacrylate (r = −0.29; *p*_adj_ = 0.016), and indolealdehyde (r = −0.29; *p*_adj_ = 0.016). Similarly, serum fTLI concentrations were correlated positively with kynurenine (r = 0.29; *p*_adj_ = 0.013) and negatively with tryptophan (r = −0.38; *p*_adj_ = 0.001), indolealdehyde (r = −0.38; *p*_adj_ = 0.001), indoleacrylate (r = −0.38; *p*_adj_ = 0.001), and indolelactate (r = −0.28; *p*_adj_ = 0.013). There were numerous statistically significant correlations among tryptophan catabolites, which are summarized (Table 5, Figure 3). The complete results of the correlation analysis are shown in the Appendix A.

## 4. Discussion

We sought to determine whether serum concentrations of tryptophan and its endogenous and exogenous (microbial) catabolites varied among cats with CIE, LGITL, and healthy controls. We hypothesized that serum concentrations of tryptophan and MICTs would be decreased and those of kynurenine would be increased in cats with CIE and LGITL compared with healthy controls. Consistent with our hypotheses, serum concentrations of tryptophan and several MICTs (indolepropionate, indolepyruvate, indolelactate, indolealdehyde, indoleacrylate) were significantly lower in cats with CIE and LGITL compared with healthy controls. Median serum concentrations of kynurenine were higher in the sera of cats with CIE and LGITL than in healthy controls, but the differences were not statistically significant. There were statistically significant correlations among serum MICTs and established serum biomarkers of gastrointestinal and pancreatic health, including cobalamin, fPLI, and fTLI. Finally, there were also numerous statistically significant correlations among tryptophan, MICTs, and endogenous tryptophan catabolites (kynurenine, kynurenate, serotonin). These findings have established strong evidence for altered tryptophan catabolism in cats with CIE and LGITL.

Our findings are consistent with previous studies that have documented decreased tryptophan and increased kynurenine in humans with IBD [23,26]. Similarly, another investigation identified decreased plasma concentrations of tryptophan in cats with CE and an inverse association between plasma tryptophan concentrations and the severity of clinical signs [46]. It is plausible that, like humans with IBD, decreased serum concentrations of tryptophan were associated with increased catabolism of tryptophan through the kynurenine pathway (KP) due to intestinal inflammation. Inflammatory cytokines including IFN-γ, IFN-α, and IL-6 induce upregulation of IDO1 activity, resulting in increased catabolism of tryptophan through the IDO1-dependent extrahepatic KP in intestinal epithelial cells and leukocytes during pro-inflammatory conditions [8,11]. In humans and rodents, the IDO1-mediated KP controls the systemic balance of kynurenine and tryptophan and its activation is known to result in decreased concentrations of tryptophan and increased concentrations of kynurenine in serum [8]. Kynurenine is an endogenous aryl hydrocarbon receptor (AhR) agonist, which regulates both innate and adaptive immune responses. Activation of the IDO1-dependent KP is associated with increased secretion of immunomodulatory cytokines IL-10 and IL-22, development of Foxp3^+^ T-regulatory cells, lymphocyte apoptosis, and T-cell suppression [11]. Thus, activation of the IDO1-dependent KP may be a counterregulatory mechanism to modulate inflammatory responses in diseased intestinal mucosal tissues [25,28]. Intestinal malabsorption due to mucosal infiltrative lesions is another likely contributor to decreased serum tryptophan concentrations in cats with CE as we have observed increased fecal concentrations of tryptophan in cats with CE (unpublished preliminary data). Follow-up investigations are needed to understand the mechanisms of altered tryptophan homeostasis, and to determine whether the serum tryptophan is a useful biomarker in cats with CE.

MICTs are considered exclusive products of microbial tryptophan catabolism and previous studies in cats with CE have identified decreased abundances of bacteria known to catabolize tryptophan into MICTs in humans and rodents, including *Bacteroides* and *Bifidobacterium* [6,7,12,17]. Thus, decreased serum concentrations of MICTs observed in cats with CIE and LGITL may be influenced by altered microbial tryptophan catabolism associated with enteric microbiota dysbiosis. MICTs modulate intestinal mucosal homeostasis by regulating inflammation and mucosal barrier function via aryl hydrocarbon (AhR) and pregnane-X (PXR) receptor-mediated expression of tight junction proteins (TJP), mucin, anti-microbial peptides (AMD), and anti-inflammatory cytokines (IL-10, IL-22) and decreased expression of pro-inflammatory cytokines (TNFα) in the enteric mucosa [20,21,25,28,29,31,47,48]. MICTs are not only potential biomarkers for EMD-associated changes in host–microbiome signaling and associated disruptions in intestinal homeostasis but are also promising therapeutic targets. Administration of probiotics containing MICT-generating species and/or dietary MICT supplementation may ameliorate mucosal inflammation and EMD in animal models of IBD [30,32,33,49]. Given the putative roles MICTs play in regulating intestinal health, it is plausible that altered tryptophan catabolism is a mechanism through which dysbiosis can contribute to the pathogenesis and/or pathophysiology of CIE and LGITL in cats. Future investigations are needed to determine whether enteric microbiota dysbiosis in CIE and LGITL is associated with altered microbial tryptophan catabolism and identify commensal microbiota that can catabolize tryptophan into MICTs in cats.

Other catabolites of tryptophan, including serotonin and tryptamine, did not differ significantly among the groups. However, serum concentrations of serotonin and tryptamine were strongly and positively correlated, and serotonin was also positively correlated with indoleacetate. Like other MICTs, indoleacetate is presumed to be an exclusive product of microbial tryptophan catabolism, whereas tryptamine and folate can also be synthesized by gut microbes. These associations suggest that enteric microbiota may influence serotonin metabolism in cats. This interpretation is supported by previous studies that have established a mechanistic role for enteric microbiota in regulating intestinal synthesis of serotonin in rodents [19,50]. Serotonin concentrations are decreased in sera, feces, and the colonic mucosa of germ-free mice (GF), and the expression of colonic tryptophan hydroxylase-1 (Tph1), the rate-limiting enzyme in serotonin synthesis, is downregulated compared with specific-pathogen-free mice (SPF) [17,19]. Conventionalization of GF mice with microbiota from SPF mice results in the restoration of serotonin concentrations in the serum and colonic mucosa, and an increased expression of Tph1, whereas antibiotic treatment recapitulates the GF state of serotonin depletion [19]. Both serotonin and tryptamine can influence GI motility and secretions via interactions with serotonin receptors [18,51]. The roles, if any, of serotonin and tryptamine in GI homeostasis and the pathophysiology of chronic enteropathies in cats are undetermined and should be targets of future investigations.

Our findings should be interpreted considering several limitations. Archived samples collected from cats with CIE and LGITL were used for this investigation. These samples were collected and stored in different locations for variable periods of time and may have been exposed to freeze–thaw cycles. The long-term stabilities of MICTs in frozen feline serum have not been specifically studied and we cannot exclude the possibility that these pre-analytic conditions may have affected the concentrations of tryptophan or its catabolites in the serum samples. Though we measured tryptophan and its catabolic derivatives against known concentrations of analytic standards (absolute quantification), the LC-MS/MS panel used has not been validated for feline serum samples. Investigations to assess the stability of analytes and to validate the assays for feline serum are currently underway. Though the diagnostic investigation of cats with CE was very similar among institutions, it was not identical. Different procedures were used to collect intestinal biopsies, including exploratory laparotomy for full-thickness biopsies and endoscopy for partial-thickness mucosal biopsies. It is possible that different histopathologic diagnoses could have been obtained if full-thickness surgical biopsies had been analyzed for all patients. Also, despite extensive testing and exclusion of ambiguous cases, it remains possible that some cats were misclassified as either CIE or LGITL. There are currently no validated criteria for the definitive differentiation of CIE and LGITL. Based on BIOMED2 guidelines in humans and a recent consensus statement from the American College of Veterinary Internal Medicine, all data, including clinical, laboratory, histopathological, immunohistochemical and clonality data, should be used to determine a final diagnosis [37,42]. This approach has been adopted in this study. It is possible these factors could have impacted the comparisons among CIE and LGITL cats; however, our findings of decreased tryptophan and MICTs were nearly identical in these groups and significantly different from healthy control cats in a manner consistent with IBD in humans and rodent models. Thus, we conclude that altered microbial tryptophan catabolism is present in both cats with CIE and LGITL, regardless of the histopathologic distinction between these disorders. Intestinal biopsies were not collected from the healthy cats, and we cannot conclude that they lacked mucosal infiltrates, which could also have impacted our results. A previous investigation identified lesions consistent with CIE and LGITL in cats lacking clinical signs of gastrointestinal dysfunction [52]. It is possible that some cats in the healthy control group had subclinical CIE or LGITL. Nonetheless, these were cats lacking clinical signs or clinicopathologic evidence of CE and we were able to demonstrate statistically significant differences in tryptophan and its catabolites when apparently healthy cats were compared with cats with active clinical signs and histopathologic lesions consistent with CIE or LGITL. Using archived samples, the authors were not able to control for the diets of the cats involved in this study. A previous study identified diet-associated differences in fecal indole concentrations, and it is possible our results could have been affected by differences in diet among cats [53]. Some cats included in this investigation had extra-intestinal comorbidities. Though we cannot exclude the possibility that the presence of these disorders affected our results, a sub-analysis did not reveal any statistically significant differences among cats with CE with respect to different extra-intestinal comorbidities (Appendix A). As chronic renal disease and hyperthyroidism are common in geriatric cats, and inflammatory disorders of the liver and pancreas are common in cats with CE, it will be important to establish the impact of these comorbidities on tryptophan catabolism in cats before their potential as biomarkers in feline CE can be determined. Owing to the use of archived samples and retrospective patient data, information about cats’ demographic data and clinical characteristics was unavailable or collected inconsistently (e.g., breed, diet, results of routine clinicopathologic tests). This prevented a complete demographic and clinical description of the cats included in this study. Finally, it must be noted that this study was not intended to investigate the clinical aspect of CIE or LGITL in cats and the clinical significance of our findings is unknown. Prospective and mechanistic studies that address these limitations are needed to confirm our findings and determine their pathophysiologic and clinical significance.

## 5. Conclusions

For the first time, and consistent with studies in humans and animal models of IBD, we have identified decreased concentrations of microbial indole catabolites of tryptophan (i.e., indolepropionate, indolepyruvate, indolelactate, indolealdehyde, and indoleacrylate) in cats with CIE and LGITL. It is plausible that decreased microbial indole catabolites of tryptophan contribute to the pathophysiology of CIE and LGITL by promoting intestinal inflammation and decreasing mucosal barrier integrity. The discovery and characterization of novel biomarkers of microbial metabolism are important to identify sub-populations of cats with chronic enteropathies in which different pathomechanisms of dysbiosis are active. In the future, these insights could be exploited to develop novel diagnostics and therapies targeting metabolic pathways in enteric microbiota that regulate intestinal mucosal health.

## Figures and Tables

**Figure 1 animals-14-00067-f001:**
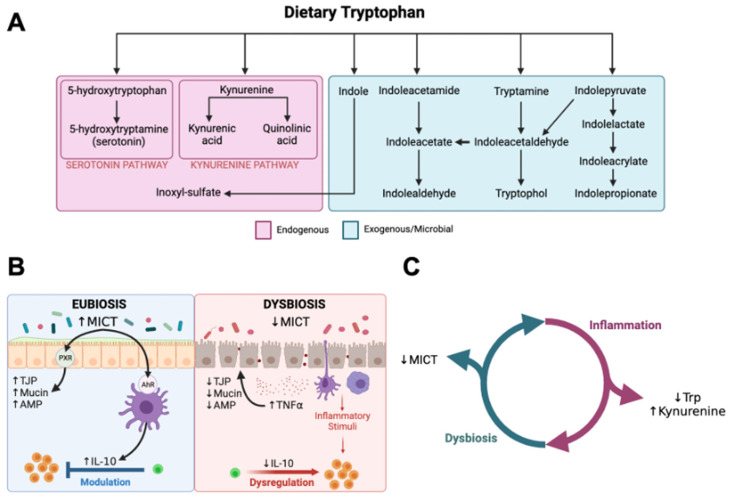
Microbial indole catabolites of tryptophan. (**A**) Arrows represent hypothetical pathways for tryptophan catabolism summarized from previous studies. Endogenous (mammalian) catabolic pathways are presented in the pink box. Gut microbial pathways are presented in the green box. (**B**) Proposed mechanisms through which microbial indole catabolites of tryptophan impact intestinal mucosal health from research in rodents and in vitro in human intestinal epithelial cell cultures. MICTs support intestinal mucosal homeostasis via interactions with aryl-hydrocarbon (AhR) and pregnane-X (PXR) receptors to increase expression of tight junction proteins (TJP), mucin, anti-microbial peptides (AMP), and IL-10. Synthesis of MICTs is reduced in dysbiosis associated with IBD and this can perpetuate mucosal inflammation and increase intestinal permeability due to decreased IL-10 and increased expression of pro-inflammatory cytokines (TNFα). (**C**) Graphical hypothesis. Mucosal inflammation in CE will increase catabolism of tryptophan via the kynurenine pathway and CE-associated dysbiosis will decrease synthesis of MICTs by gut microbes in cats with CE. Figures created with BioRender.com. (**B**) was adapted from “Immune Response in Inflammatory Bowel Disease”, by BioRender.com (2023), retrieved from https://app.biorender.com/biorender-templates, accessed on 14 December 2023.

**Figure 2 animals-14-00067-f002:**
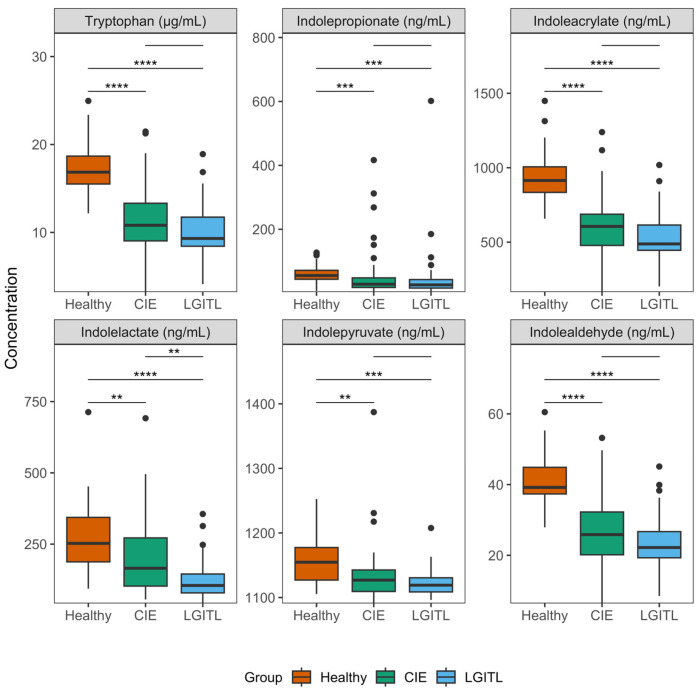
Serum concentrations of significantly variable tryptophan catabolites. Boxplots were drawn for each variable that differed significantly among the groups in the overall Kruskal–Wallis tests. The upper and lower boundaries of the box represent the 25th and 75th percentiles and the horizontal line represents the median. The whiskers represent the maximum and minimum values below and above the upper (75th percentile + IQR) and lower (and 25th percentile—IQR) fences, respectively. Bars denote statistically significant group-wise comparisons from post hoc Dunn’s tests and the asterisks correspond to the resulting adjusted *p*-values (*p*_adj_). **, *p*_adj_ < 0.01; ***, *p*_adj_ < 0.001, ****, *p*_adj_ < 0.0001. CIE, chronic inflammatory enteropathy; LGITL, low-grade intestinal T-cell lymphoma.

**Figure 3 animals-14-00067-f003:**
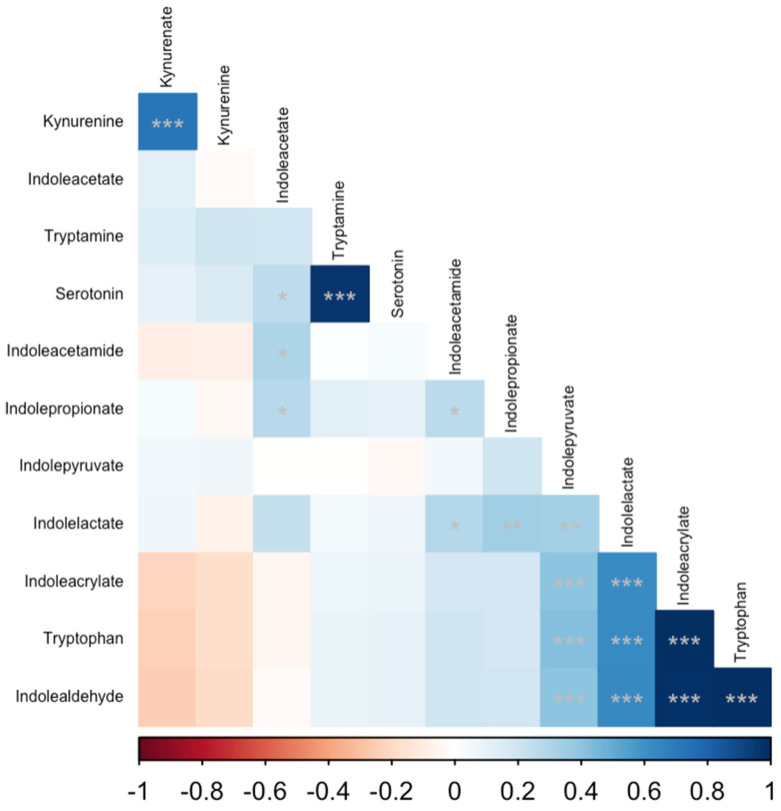
Correlation matrix of serum tryptophan catabolites. Spearman’s rank correlation coefficients were calculated for all pairwise combinations. The intensity of the colored cells is proportional to Spearman’s rho, with blue cells indicating a positive correlation and red cells indicating a negative correlation. The matrix is ordered by the first principal component of the correlation matrix. Each cell is annotated with asterisks corresponding to the *p*-value for a given correlation: * *p* < 0.05; ** *p* < 0.01; *** *p* < 0.001.

**Table 1 animals-14-00067-t001:** Demographic and clinical data. The *p*-values were generated from Kruskal–Wallis tests (age, BCS, cobalamin, folate, fPLI, fTLI) or Chi-Square tests (sex). Post hoc Dunn’s tests were performed for numeric variables that were significantly different (*p* < 0.05) among the groups in the Kruskal–Wallis test. Groups with the same superscript did not differ significantly, whereas those with different superscripts were significantly different in post hoc Dunn’s tests (*p*_adj_ < 0.05). CIE, chronic inflammatory enteropathy; LGITL, low-grade intestinal T-cell lymphoma. The upper and lower limits of detection for serum cobalamin were 150 ng/L and 1000 ng/L, respectively.

	Healthy (*n* = 26)	CIE (*n* = 44)	LGITL (*n* = 31)	*p*-Value
Age (years)				0.010
Median	10.0 ^a^	10.5 ^a,b^	12.0 ^b^	
IQR	8.0–11.0	7.0–13.0	8.0–13.0	
Min–Max	3–14	2–17	7–15	
Sex				0.85
Spayed Female	15 (57.7%)	23 (52.3%)	18 (58.1%)	
Neutered Male	11 (42.3%)	21 (47.7%)	13 (41.9%)	
BCS (1–9)				<0.001
Median	6 ^a^	5 ^b^	4 ^b^	
IQR	5–8	4–5	4–5	
Min–Max	5–9	3–7	1–9	
Cobalamin (ng/L)				<0.001
Median	1000.0 ^a^	743.0 ^b^	615.0 ^b^	
IQR	826.3–1000.0	378.8–891.3	209.0–906.5	
Min–Max	311.0–1000.0	150.0–1000.0	150.0–1000.0	
Folate (µg/L)				0.66
Median	15.8	16.1	16.0	
IQR	14.1–17.9	10.7–19.7	13.2–21.2	
Min–Max	11.0–21.5	5.2–38.0	9.2–48.0	
fPLI (µg/L)				0.045
Median	2.0 ^a^	2.0 ^a^	2.7 ^a^	
IQR	1.4–2.5	1.3–2.98	1.5–5.98	
Min–Max	0.9–3.2	0.5–21.6	1.0–51.0	
fTLI (µg/L)				<0.001
Median	32.6 ^a^	32.2 ^a^	56.8 ^b^	
IQR	23.1–39.0	27.8–47.5	38.5–84.4	
Min–Max	15.9–63.7	14.3–231.0	15.4–243.7	

**Table 2 animals-14-00067-t002:** Serum concentrations of tryptophan catabolites. The *p*-values were generated from Kruskal–Wallis tests and post hoc Dunn’s tests were performed for variables that were significantly different (*p* < 0.05) among the groups. In rows containing metabolites that were significantly different, groups with the same superscript indicate variables that did not differ significantly, whereas those with different superscripts indicate variables that were significantly different in post hoc Dunn’s tests (*p*_adj_ < 0.05). CIE, chronic inflammatory enteropathy; LGITL, low-grade intestinal T-cell lymphoma.

	Healthy (*n* = 26)	CIE (*n* = 44)	LGITL (*n* = 31)	Kruskal–Wallis *p*-Value
Tryptophan (µg/mL)				<0.001
Median	16.9 ^a^	10.8 ^b^	9.3 ^b^	
IQR	15.5–18.7	9.0–13.3	8.4–11.7	
Min–Max	12.2–25.0	3.2–21.5	4.1–18.9	
Kynurenine (µg/mL)				0.23
Median	1.04	1.14	1.26	
IQR	0.888–1.237	0.843–1.57	1.01–1.53	
Min–Max	085–1.94	0.513–9.81	0.601–2.52	
Kynurenate (ng/mL)				0.63
Median	8.9	10.3	9.0	
IQR	7.9–10.9	6.5–14.9	6.6–13.8	
Min–Max	4.6–22.7	3.7–136.2	4.6–37.3	
Serotonin (ng/mL)				0.15
Median	888.0	760.9	954.2	
IQR	677.8–1160.0	516.0–1038.8	759.5–1235.9	
Min–Max	293.9–1439.3	12.8–3214.1	13.4–2158.5	
Indoleacetate (ng/mL)				0.21
Median	182.8	267.3	210.0	
IQR	139.9–300.4	172.9–420.0	95.7–418.1	
Min–Max	62.7–980.1	22.1–3219.3	51.8–1946.7	
Indolepropionate (ng/mL)				<0.001
Median	55.4 ^a^	28.2 ^b^	26.1 ^b^	
IQR	43.7–71.5	18.0–48.0	16.1–42.7	
Min–Max	7.2–127.0	3.9–416.5	3.6–601.7	
Indoleacetamide (ng/mL)				0.034
Median	1.4 ^a^	1.4 ^a^	1.4 ^a^	
IQR	1.4–1.4	1.4–1.4	1.4–1.4	
Min–Max	1.3–2.0	1.3–2.1	1.3–1.5	
Indoleacrylate (ng/mL)				<0.001
Median	914.2 ^a^	606.0 ^b^	488.3 ^b^	
IQR	834.5–1006.2	478.3–687.8	445.5–615.3	
Min–Max	657.5–1448.8	165.0–1239.3	202.7–1018.4	
Indolelactate (ng/mL)				<0.001
Median	252.7 ^a^	165.5 ^b^	105.0 ^c^	
IQR	187.8–343.8	102.9–271.8	79.0–145.2	
Min–Max	93.8–713.1	56.0–691.7	42.1–355.6	
Indolepyruvate (ng/mL)				0.001
Median	1154.7 ^a^	1127.0 ^b^	1119.1 ^b^	
IQR	1127.1–1177.5	1109.4–1142.7	1108.7–1130.6	
Min–Max	1105.4–1252.6	1090.9–1387.2	1096.3–1207.7	
Indolealdehyde (ng/mL)				<0.001
Median	39.2 ^a^	25.9 ^b^	22.2 ^b^	
IQR	37.4–44.9	20.2–32.3	19.3–26.7	
Min–Max	27.9–60.5	6.4–53.2	8.5–45.1	
Tryptamine (ng/mL)				0.160
Median	3.5	3.4	3.6	
IQR	3.2–4.0	3.0–3.7	3.3–3.9	
Min–Max	2.6–4.2	0.0–6.3	2.4–4.8	

**Table 3 animals-14-00067-t003:** Results of post hoc Dunn’s tests. The Z-statistics and adjusted *p*-values (*p*_adj_) from Dunn’s tests are shown for group-wise contrasts of every variable that differed significantly among the groups in the overall Kruskal–Wallis tests. CIE, chronic inflammatory enteropathy; LGITL, low-grade intestinal T-cell lymphoma.

Variable	Contrast	Dunn’s Z Statistic	*p*_adj_ Value
Tryptophan	Healthy vs. CIE	−4.96	<0.001
	Healthy vs. LGITL	−5.83	<0.001
	CIE vs. LGITL	−1.37	0.17
Indolepropionate	Healthy vs. CIE	−3.37	0.001
	Healthy vs. LGITL	−3.60	<0.001
	CIE vs. LGITL	−0.53	0.60
Indolepyruvate	Healthy vs. CIE	−3.14	0.003
	Healthy vs. LGITL	−3.43	0.002
	CIE vs. LGITL	−0.58	0.56
Indolelactate	Healthy vs. CIE	−2.60	0.009
	Healthy vs. LGITL	−4.87	<0.001
	CIE vs. LGITL	−2.77	0.008
Indolealdehyde	Healthy vs. CIE	−4.88	<0.001
	Healthy vs. LGITL	−5.74	<0.001
	CIE vs. LGITL	−1.36	0.17
Indoleacrylate	Healthy vs. CIE	−5.08	<0.001
	Healthy vs. LGITL	−5.93	<0.001
	CIE vs. LGITL	−1.36	0.17

**Table 4 animals-14-00067-t004:** Correlations among tryptophan catabolites and established gastrointestinal and pancreatic biomarkers in serum. Spearman’s rank correlation coefficients (rho) were calculated for combinations of tryptophan catabolites and clinical gastrointestinal biomarkers. *p*-values are adjusted for multiple comparisons (*p*_adj_) using the Benjamini–Hochberg method to control for false discovery. Shown here are statistically significant correlations (*p*_adj_ < 0.05).

Clinical Variable	Tryptophan Catabolite	Spearman’s Rho	*p* _adj_
Cobalamin	Indolealdehyde	0.39	<0.001
	Tryptophan	0.39	0.001
	Indoleacrylate	0.37	0.001
	Indolepyruvate	0.31	0.007
	Indolelactate	0.27	0.02
	Kynurenine	−0.25	0.03
fPLI	Kynurenine	0.28	0.016
	Tryptophan	−0.28	0.016
	Indoleacrylate	−0.29	0.016
	Indolealdehyde	−0.29	0.016
fTLI	Kynurenine	0.29	0.013
	Indolelactate	−0.28	0.013
	Indoleacrylate	−0.38	0.001
	Indolealdehyde	−0.38	0.001
	Tryptophan	−0.38	0.001

**Table 5 animals-14-00067-t005:** Correlations among tryptophan catabolites. Spearman’s rank correlation coefficients (rho) were calculated for pairwise combinations of tryptophan catabolites. *p*-values are adjusted for multiple comparisons (*p*_adj_) using the Benjamini–Hochberg method to control for false discovery. Shown here are statistically significant correlations (*p*_adj_ < 0.05).

		Spearman’s Rho	*p* _adj_
Tryptamine	Serotonin	0.98	<0.001
Tryptophan	Indolealdehyde	0.99	<0.001
	Indoleacrylate	0.99	<0.001
	Indolelactate	0.64	<0.001
	Indolepyruvate	0.42	<0.001
Serotonin	Indoleacetate	0.26	0.034
Kynurenate	Kynurenine	0.72	<0.001
Indolepropionate	Indoleacetate	0.27	0.026
Indolepyruvate	Indoleacrylate	0.41	<0.001
	Indolelactate	0.34	0.003
Indolelactate	Indoleacrylate	0.62	<0.001
	Indolepropionate	0.34	0.0023
	Indoleacetamide	0.28	0.02
Indolealdehyde	Indoleacrylate	0.99	<0.001
	Indolelactate	0.65	<0.001
	Indolepyruvate	0.41	<0.001
Indoleacetamide	Indoleacetate	0.3	0.01
	Indolepropionate	0.27	0.027

## Data Availability

All data and R code necessary to replicate this analysis are located in a GitHub repository (https://github.com/pcbarko/FCE_Microbial_Indole_Catabolites; accessed on 14 December 2023).

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
