# Peer review of "Chronic Inflammatory Enteropathy and Low-Grade Intestinal T-Cell Lymphoma Are Associated with Altered Microbial Tryptophan Catabolism in Cats"

_animals, 2023, doi:10.3390/ani14010067_

Round 1

Reviewer 1 Report

Comments and Suggestions for Authors

Comments to Chronic inflammatory enteropathy and low-grade intestinal T cell lymphoma are associated with altered microbial tryptophan catabolism in cats. Manuscript ID: animals-2705734

 The research is related to tryptophan metabolites in cats with inflammatory enteropathy and low-grade lymphoma. It is a well written paper with clear objectives. However, I have a few comments to make.

The introduction is quite complete, and I have no further comments on it.

In the materials and methods section, it is mandatory to divide each procedure under subheadings: number and characteristics of the animals included in the assays data (demographic data appeared insufficient as breed or diet were not included) , inclusion and exclusion criteria, blood tests, handling of serum samples, etc. The same applies for the LCMS.

Results

233, page 5, Table 1

It is not understood why the authors chose to present the data for serum concentrations of endogenous tryptophan catabolites in table and serum concentrations of significantly variable tryptophan catabolites in figure. Please explain why this selection was made.

I was most puzzled by the absence of data on leukocyte values in these animals, as well as the clinical description of the animals involved in the trial.

Discussion

The major concerns about this work are explained by the authors in the discussion section. The interpretation of the results with serum samples and histopathology from various institutions under different management is difficult. Also, the different types of diet definitely have an impact on these results (“the authors were not able to control for the diets of the cats involved in this study. It is possible our results could have been affected by differences in dietary tryptophan intake”). Even though statistical differences may have been found, it is challenging to get accurate conclusions in this respect.

The authors should discuss about tryptophan metabolites in those cases of faecal transplantation, a palliative measure for IBD.  The issue of microbiome analysis should be mentioned as a future research. A discussion about the analysis of tryptophan catabolites into daily clinical practice is needed. Can these biomarkers really be applied in routine examinations and do they give more useful information for treatment of IBD or LGL?

The gut tissue has become a Pandora's Box. Analysis of the microbiome and the species involved in inflammatory events is needed to corroborate the results obtained here.

Please change the names of supplementary files to be more descriptive.

I must make an additional comment. The responsibility of the authors is submitting all the material associated with the manuscript at the same time. I asked the editorial office for supplementary files S1 and S2, which reached me several days after receipt of the manuscript and caused the delay of this evaluation.  I also searched for the data associated with the research in the repository and did not find it. Therefore, authors cannot refer to a repository that is not visible to the public or in which data does not exist. This should therefore be changed or be removed from the manuscript.

Author Response

Reviewer #1

The research is related to tryptophan metabolites in cats with inflammatory enteropathy and low-grade lymphoma. It is a well written paper with clear objectives. However, I have a few comments to make.

Response from Authors: Thank you for taking the time to review this manuscript and for your compliments related to its composition and the clarity of the objectives.

The introduction is quite complete, and I have no further comments on it.

Response from Authors: Thank you.

In the materials and methods section, it is mandatory to divide each procedure under subheadings: number and characteristics of the animals included in the assays data (demographic data appeared insufficient as breed or diet were not included), inclusion and exclusion criteria, blood tests, handling of serum samples, etc. The same applies for the LCMS.

Response from Authors: Thank you for this comment. We have revised the “Materials and Methods” section to include section sub-headings, which we agree will improve readability of the manuscript. Owing to the use of archived samples and retrospective data, some demographic information was either unavailable or collected in an inconsistent manner, including the cat’s breeds and dietary information. We mention this as a limitation in our “Discussion” section [lines].

Results

233, page 5, Table 1

It is not understood why the authors chose to present the data for serum concentrations of endogenous tryptophan catabolites in table and serum concentrations of significantly variable tryptophan catabolites in figure. Please explain why this selection was made.

Response from Authors: In deciding how to display our findings, we wanted to be as transparent as possible in presenting the summary statistics for all variables. Thus, we have included results from all analytes in table form so readers can understand the measures of central tendency (medians) and variation (interquartile ranges and min-max) for all variables. Regarding graphical representation, we elected to show plots of analytes that varied significantly in order to emphasize the most important results. If we had elected to plot all analytes, including those that did not vary significantly, there would have been a total of 12 individual box plots. This would have taken up a lot of space in the manuscript and would not have provided any additional information to the reader (the medians and IQRs in boxplots are also noted in the tables). By omitting plots from the analytes that did not vary significantly, the authors were not trying to conceal anything. Simply put, we were trying to present our results in the simplest and most efficient manner while providing sufficient information about the non-significant findings in the tables so that readers can understand the how those variables compared among groups. To improve readability we have combined the two tables containing the tryptophan catabolite data into a single table in the revised manuscript. In the interest of maximum transparency, we have also included boxplots for all analytes in the supplemental data (S1 File).

I was most puzzled by the absence of data on leukocyte values in these animals, as well as the clinical description of the animals involved in the trial.

Response from Authors: We understand your concerns related to the lack of clinical information about the cats included in this study. This was a proof-of-concept investigation to determine if altered microbial tryptophan catabolism is present in cats with chronic enteropathies. The samples had been archived from previous studies, and residual serum from diagnostics had been performed at different institutions. Thus, clinical data were not collected in a coordinated or consistent fashion. Samples for routine clinical pathology (complete blood counts, serum chemistries) had been collected at different timepoints in relation to the samples used for tryptophan catabolite measurements, and clinicopathologic assessments were performed on different instruments with different reference intervals. Thus, we do not consider these results to be comparable for the purposes of a clinical investigation. Additionally, it was not our objective to describe clinical features of feline chronic enteropathy and our study was not designed to accomplish that. We agree that additional clinical data could help place our findings in a clinical context. However, we do not think it is valid to present comparisons of variables that were not collected in a consistent manner. Finally, we agree that prospective investigations are absolutely required to confirm our results and place them in a clinically relevant context. We acknowledge this openly in our discussion.

Discussion

The major concerns about this work are explained by the authors in the discussion section. The interpretation of the results with serum samples and histopathology from various institutions under different management is difficult. Also, the different types of diet definitely have an impact on these results (“the authors were not able to control for the diets of the cats involved in this study. It is possible our results could have been affected by differences in dietary tryptophan intake”). Even though statistical differences may have been found, it is challenging to get accurate conclusions in this respect.

Response from Authors: As we state in the “Discussion” section, we fully acknowledge these limitations. As stated in the manuscript, our findings must be replicated in rigorously controlled, prospective investigations.

The authors should discuss about tryptophan metabolites in those cases of faecal transplantation, a palliative measure for IBD.  The issue of microbiome analysis should be mentioned as a future research. A discussion about the analysis of tryptophan catabolites into daily clinical practice is needed. Can these biomarkers really be applied in routine examinations and do they give more useful information for treatment of IBD or LGL?

Response from Authors: Respectfully, speculation of the effect of fecal microbiota transplantation, or other therapies, on microbial tryptophan catabolism is beyond the scope of this investigation. Additionally, fecal microbiota transplantation is not an established therapy for feline chronic enteropathies and has not been evaluated for safety or efficacy for the treatment of CIE or LGITL in controlled clinical trials. Future, prospective investigations are required to determine whether treatment of feline chronic enteropathy is associated with changes in serum concentrations of tryptophan catabolites.

Regarding application of our results to “daily clinical practice,” the authors cannot make any claims to the clinical significance of our findings. Our study was not designed to assess any clinical aspect of microbial tryptophan catabolism in cats. We address the potential for these findings to generate clinically meaningful insights in future investigations in our “Conclusion” section. If our findings are confirmed and shown to have clinical relevance in prospective investigations, it is possible that assays for microbial indole catabolites of tryptophan could be incorporated into diagnostic schemes, however much more research is required before this is possible.

The gut tissue has become a Pandora's Box. Analysis of the microbiome and the species involved in inflammatory events is needed to corroborate the results obtained here. 

Response from Authors: We agree. This was a proof-of-concept study and future investigations are needed to provide pathophysiological context, especially with respect to the nexus of enteric microbiota dysbiosis, microbial metabolism, and mucosal inflammation. Our study was not intended to provide this level of mechanistic detail.

Please change the names of supplementary files to be more descriptive.

Response from Authors: The requested changes have been made.

I must make an additional comment. The responsibility of the authors is submitting all the material associated with the manuscript at the same time. I asked the editorial office for supplementary files S1 and S2, which reached me several days after receipt of the manuscript and caused the delay of this evaluation.  I also searched for the data associated with the research in the repository and did not find it. Therefore, authors cannot refer to a repository that is not visible to the public or in which data does not exist. This should therefore be changed or be removed from the manuscript.

Response from Authors: The authors acknowledge that an error was made when the files were uploaded for this manuscript. Due to an accidental technical mistake, the supplemental data was inadvertently omitted when the manuscript was submitted. We regret this mistake and thank the reviewer for their efforts to obtain and review the supplemental information. Regarding the data and code repository, it has now been made public and can be accessed by the reviewer at the following url: https://github.com/pcbarko/FCE_Microbial_Indole_Catabolites.  

Reviewer 2 Report

Comments and Suggestions for Authors

1.  In this fascinating research study, titled "Chronic inflammatory enteropathy and low-grade intestinal T- cell lymphoma are associated with altered microbial tryptophan catabolism in cats".Study examined serum from healthy cats and cats with CIE and LGITL, Cats with CIE and LGITL had decreased serum concentrations of tryptophan and microbial indole catabolites of tryptophan. Do other types of enteritis cause similar results?

2. L181-L183,.What is the purpose of these tests?

3. Table 1 and Table 2 are in different formats. There should be no Spaces in the first column of Table 1

4. L278 is in italics, in the same format as the other tables.

5. L306 is in italics, in the same format as the other figures.

6. Adjust the position of the tables and figures on the page so that they are consistent, which will be more comfortable

Author Response

Reviewer #2

In this fascinating research study, titled "Chronic inflammatory enteropathy and low-grade intestinal T- cell lymphoma are associated with altered microbial tryptophan catabolism in cats".Study examined serum from healthy cats and cats with CIE and LGITL, Cats with CIE and LGITL had decreased serum concentrations of tryptophan and microbial indole catabolites of tryptophan. Do other types of enteritis cause similar results?

Response from Authors: Thank you for your efforts to review this manuscript and for your kind remarks. We have not investigated whether other chronic enteropathies or other forms of enteritis are associated with altered microbial tryptophan catabolism. Future studies are needed to understand this.

L181-L183 What is the purpose of these tests?

Response from Authors: These tests are needed to identify clinically significant diseases that would result in exclusion of cats from the healthy control group. Cats with clinically significant systemic, gastrointestinal, and pancreatic diseases (chronic renal disease, hepatobiliary disease, pancreatitis) could affected our results. Thus, samples from healthy cats used in the control group were obtained from unrelated prospective studies in which the investigators ruled out confounding disorders using routine clinicopathologic screening tests. Complete blood count, serum chemistry profile, and serum concentrations of total T4 are routine clinicopathologic tests to exclude metabolic, endocrine, and systemic diseases. Serum concentrations of cobalamin, folate, feline pancreatic lipase immunoreactivity, and feline trypsin-like immunoreactivity are routine biomarkers used to screen for gastrointestinal and pancreatic diseases.

Table 1 and Table 2 are in different formats. There should be no Spaces in the first column of Table 1

Response from Authors: These two tables have been combined and the formatting has been corrected.

L278 is in italics, in the same format as the other tables.

Response from Authors: This has been corrected.

L306 is in italics, in the same format as the other figures.

Response from Authors: This has been corrected.

Adjust the position of the tables and figures on the page so that they are consistent, which will be more comfortable

Response from Authors: We have used the formatting template provided by the Journal. We believe this will be fixed in the final proofs prior to publication, should our manuscript be accepted.

Reviewer 3 Report

Comments and Suggestions for Authors

This study is very interesting and constitutes progress in our understanding of the microbiota-host dialogue during digestive disease.

Basically, the lack of access to dietary data constitutes a real limitation. This is discussed at the end of the article. When it comes to a comparison between healthy cats and potentially seriously ill cats, it would be interesting to have at least a comparison of appetite and food intake between the groups. Cats with lymphoma may be anorexic. The supply of trytophan would then be seriously disrupted.

It would be particularly interesting to have an analysis of the composition of the intestinal microbiota in order to look for possible bacterial signatures (to be completed or discussed depending on the possibilities).

L.400-401: to the extent that we cannot affirm that the dietary tryptophan intake is similar between the groups, is it really possible to affirm that the reductions in tryptophan catabolites observed are linked to the disease ? and not to one of its consequences (anorexia)? or intestinal malabsorption?

L. 402. Increased kynurenine is not documented in the present study

L. 406-408: Moderate this statement

L. 429-431: Moderate this statement

L. 468-474: not sure that the analogy with the dog is useful for understanding the idea of sub-populations. The article is overall quite long.

L. 531-533: same idea as the previous sentence. Merge the 2 sentences.

L. 661-662: format the reference

Author Response

Reviewer #3

This study is very interesting and constitutes progress in our understanding of the microbiota-host dialogue during digestive disease.

Response from Authors: Thank you for reviewing our manuscript and for your kind comments.

Basically, the lack of access to dietary data constitutes a real limitation. This is discussed at the end of the article. When it comes to a comparison between healthy cats and potentially seriously ill cats, it would be interesting to have at least a comparison of appetite and food intake between the groups. Cats with lymphoma may be anorexic. The supply of trytophan would then be seriously disrupted.

Response from Authors: We agree that additional dietary and clinical data would provide important context for our findings. Unfortunately, this information was not available for all cats included in this investigation and was not collected in a consistent manner. These limitations are related to our use of archived samples from investigations at different institutions and residual clinical samples. Our proof-of-concept study should inform future investigations in which these important variables can be more rigorously controlled and characterized.

It would be particularly interesting to have an analysis of the composition of the intestinal microbiota in order to look for possible bacterial signatures (to be completed or discussed depending on the possibilities).

Response from Authors: We agree that this is an important future direction and we have addressed this in our “Discussion” section.

L.400-401: to the extent that we cannot affirm that the dietary tryptophan intake is similar between the groups, is it really possible to affirm that the reductions in tryptophan catabolites observed are linked to the disease ? and not to one of its consequences (anorexia)? or intestinal malabsorption?

Response from Authors: As we state in the “Discussion” section, we cannot definitively conclude that decreased serum MICT concentrations are caused by dysbiosis associated with chronic enteropathy. We consider that the most likely cause, given similar findings in humans and rodent models of inflammatory bowel diseases. However, we cannot exclude intestinal malabsorption or dietary differences as contributing factors.

  1. 402. Increased kynurenine is not documented in the present study

Response from Authors: Median kynurenine concentrations were increased in the CIE and LGITL groups, compared with the healthy controls, but these results were not statistically significant.

406-408: Moderate this statement

 Response from Authors: This statement has been moderated as requested.

429-431: Moderate this statement

 Response from Authors: This statement has been moderated as requested.

468-474: not sure that the analogy with the dog is useful for understanding the idea of sub-populations. The article is overall quite long.

Response from Authors: This information has been removed.

531-533: same idea as the previous sentence. Merge the 2 sentences.

Response from Authors: Thank you for identifying this redundancy. We have merged the two sentences as requested.

661-662: format the reference

Response from Authors: The reference has been formatted.

Round 2

Reviewer 1 Report

Comments and Suggestions for Authors

Dear authors,

I have no further comments to make about this revised version of the manuscript.